# Outpatient Spine Procedures in Poland: Clinical Outcomes, Safety, Complications, and Technical Insights into an Ambulatory Spine Surgery Center

**DOI:** 10.3390/healthcare11222944

**Published:** 2023-11-10

**Authors:** Kajetan Latka, Waldemar Kolodziej, Kacper Domisiewicz, Dawid Pawus, Tomasz Olbrycht, Marcin Niedzwiecki, Artur Zaczynski, Dariusz Latka

**Affiliations:** 1Department of Neurosurgery, The St. Hedwig’s Regional Specialist Hospital, 45-221 Opole, Poland; 2Department of Neurosurgery, Institute of Medical Sciences, University of Opole, 45-040 Opole, Poland; waldemar.kolodziej@usk.opole.pl (W.K.); tomasz.olbrycht@usk.opole.pl (T.O.); dariusz.latka@usk.opole.pl (D.L.); 3Center for Minimally Invasive Spine and Peripheral Nerve Surgery neurochirurg.opole.pl, 45-064 Opole, Poland; kacperdomisiewicz@gmail.com (K.D.); marcin.niedzwiecki@cskmswia.gov.pl (M.N.); 4Faculty of Electrical Engineering, Automatic Control and Informatics, Opole University of Technology, 45-758 Opole, Poland; dawid.pawus@o2.pl; 5Department of Neurosurgery, The National Institute of Medicine of the Ministry of Internal Affairs and Administration, 02-507 Warsaw, Poland; artur.zaczynski@cskmswia.gov.pl

**Keywords:** ambulatory surgery, spine endoscopy, minimal-invasive spine, outpatient, complications

## Abstract

Purpose: This study evaluated the safety and efficacy of spine procedures performed in an ambulatory spine surgery unit in Poland. Patients and Methods: We conducted a retrospective analysis of 318 patients who underwent ambulatory spine surgery between 2018 and 2021, with procedures including microdiscectomy (MLD), anterior cervical discectomy and fusion (ACDF), endoscopic interbody fusion (endoLIF), posterior endoscopic cervical discectomy (PECD), interlaminar endoscopic lumbar discectomy IELD, and transforaminal endoscopic lumbar discectomy (TELD). Patient data were analyzed for pre-operative and post-operative visual analog scale (VAS) scores. Results: The findings indicated that outpatient techniques were safe and effective, with a 2.83% complication rate. All procedures significantly improved VAS scores under short-term observation, and core outcome measurement index (COMI) scores under long-term observation. Conclusions: Ambulatory spine surgery represents a relatively new approach in Poland, with only a select few centers currently offering this type of service. Outpatient spine surgery is a safe, effective, and cost-effective option for patients requiring basic spine surgeries.

## 1. Introduction

Spine surgeries are among the most highly priced surgical procedures in both the United States and Europe [1]. Due to the rapid development of medical technologies, especially in terms of diagnostic visualization methods, surgical access systems, spinal implants, and anesthesia methods, these procedures are becoming less invasive [2]. As a result, they are associated with smaller wounds [3], less blood loss [4], less traumatization of paraspinal muscles [5], and, according to numerous reports, with lower risk of complications [6]. Modern techniques of general anesthesia, like total intravenous anesthesia (TIVA), combined with regional methods such as epidural anesthesia or dorsal extensor muscle blockade, and local anesthesia, have significantly reduced the requirement for several days of hospitalization. This extended stay was a standard just a few years ago [7].

In countries with a liberal, free-market health care model, such as the United States [8] or South Korea [9], more surgical procedures are performed in an outpatient manner. According to some analyses, the existing rules within the Polish health system may inadvertently incentivize healthcare entities to prolong hospitalization times for the sake of reimbursement [10]. Due to the extremely long waiting lists for patients requiring spinal surgery, the private medical sector based on free market principles has started to emerge in Poland. The lack of strict regulations regarding the duration of hospitalization has permitted the use of spinal surgery not only in hospitals but also in outpatient surgical centers [11].

The aim of this retrospective study was to assess the effectiveness and safety of ambulatory spine surgery based on an analysis of 300 cases from our ambulatory practice over the past three years. By examining pre-operative and post-operative data using standardized tools, like the VAS and COMI, we sought to understand the clinical improvements achieved. We discussed the safety and efficacy of this approach, and the potential benefits it can provide to both patients and the healthcare system. This article aimed to increase awareness of the advantages of ambulatory spine surgery and encourage its adoption in the Polish healthcare system.

## 2. Materials and Methods

This article presented a retrospective analysis of spine surgeries conducted in our ambulatory center between 2018 and 2021. We collected data on over 300 outpatient procedures, which included microdiscectomies, endoscopic discectomies, cervical discectomies with fusion, endoscopic interbody fusion with percutaneous screw fixation, and posterior endoscopic cervical discectomies. As a standard practice in our center, data were consistently collected from patients as a routine evaluation of treatment outcomes. The primary goal of this study was to gauge the safety, efficacy, and levels of patient satisfaction related to outpatient spine surgeries in an ambulatory environment. The analysis and scrutiny of these procedures took place in 2022.

Our evaluation extended to counting and categorizing surgeries, gauging their durations, and appraising patient conditions utilizing two specific metrics: the core outcome measurement index (COMI) for the back and the visual analog scale (VAS) score. Furthermore, an intrinsic component of our analysis focused on the registration of complications experienced by patients.

Intra-operative complications, such as durotomy, were immediately documented and integrated into the surgical description. In contrast, delayed complications, like infections, were systematically recorded during subsequent follow-up visits. This layered approach ensured comprehensive tracking of both immediate and deferred issues, painting a holistic picture of patient outcomes. We rigorously analyze complications every six months in a collaborative setting, involving our entire medical team in discussions, deliberations, and decision-making processes to enhance our practices and protocols.

The insights derived from this study have the potential to profoundly influence clinical judgments and the broader landscape of health policies.

### 2.1. Ethical Considerations

This study was conducted in strict adherence to ethical guidelines for human subjects’ research. Prior to the commencement of the study, ethical approval was obtained from the Bioethics Committee at the Opole Chamber of Medicine. The committee reviewed and approved the study’s protocol, ensuring that it met ethical standards in the protection and respect for the rights and welfare of the participants involved. All patients provided informed consent to participate in this study, and their identities have been anonymized and de-identified to maintain confidentiality and privacy. This study was performed following the ethical standards laid down in the 1964 Declaration of Helsinki and its later amendments.

### 2.2. Ambulatory Spine Center

All procedures reported in this study were performed in a surgical outpatient center that was specifically designed for ambulatory spine surgery. Our center consists of a reception desk with a waiting room, a surgical treatment room, a two-bed post-operative observation room with a patient toilet, and small social facilities. This facility is located on the 3rd floor of a modern building in the city center, with consultation rooms located in a separate space on the same floor. The nearest neurosurgery department and emergency center are located within a 5–10 min drive. The area of the facility for direct patient care is approximately 85 square meters.

While our center’s treatment room is not legally classified as a surgical operating room, it has been registered and functions as a treatment room. Importantly, it is equipped with state-of-the-art medical equipment that is comparable to, or even superior to, what is available in traditional hospital operating rooms (Figure 1). Notably, this facility has been authorized by the Regional Epidemiological Station to perform specific procedures related to minimally invasive spine surgery.

Our treatment room boasts state-of-the-art medical equipment, including an endoscopic column (Joimax, Karlsruhe, Germany), an intra-operative microscope (Zeiss OPMI, Oberkochen, Germany), a radiolucent carbon table (Famed, Zywiec, Poland), a C-arm (Ziehm Solo, Nurnberg, Germany), ultrasound (Hitachi Aloka, Tokyo, Japan), ECG, infusion pumps, and a general anesthesia unit (Dräger, Lubeck, Germany). This suite of equipment, specially curated for our procedures, is sometimes more specialized or represents newer technology than one might find in many conventional hospital operating rooms. While our treatment room may not be officially classified as a surgical operating room, its design and equipment align with the stringent standards set out in the infection control and environmental management guidelines. Adhering to the protocols developed in collaboration with the regional sanitary and epidemiological station, this space undergoes regular disinfection. Our staff members are not only trained to maintain strict hand hygiene but also to correctly utilize personal protective equipment. Moreover, our hygiene procedures are periodically reviewed and monitored by the regional sanitary and epidemiological station to ensure continuous adherence and safety.

### 2.3. Patient Selection

According to the guidelines of the Polish Society of Spine Surgery, patients eligible for the procedures were those exhibiting radicular symptoms correlating with MRI findings and persisting for more than 6 weeks [12,13]. Patients who did not meet this criterion were disqualified. Only patients classified under ASA 1 and ASA 2 were routinely considered suitable. Patients categorized under ASA 3 were conditionally qualified: they were considered if their categorization was due to an optimally treated chronic illness, and they were in their best possible health condition. Any medical condition that precluded safe anesthesia according to the ASA criteria resulted in disqualification [14]. Patients who were unable to receive care on their first night post-procedure were also excluded, although this was a rare occurrence in our patient cohort. Those who came from distant parts of Poland were qualified provided they spent the night in our city at a nearby hotel.

### 2.4. Pre-Operative Management and Patient Education

In line with the Joint Commission International Accreditation Standards for Ambulatory Care, our center places a premium on comprehensive patient care. We employed a qualified patient consultant whose primary responsibility is to educate the patient prior to the procedure and ensure seamless communication between the patient and our medical team. Beyond face-to-face interactions, patients are provided access to a dedicated webpage, enriched with informational and educational videos encompassing the range of surgeries and anesthesia techniques we offer. As a standard practice, prior to any procedure, patients undergo a fundamental blood test package. Depending on specific cases and requirements, additional tests or consultations with specialists may be necessitated. Anesthetic premedication is facilitated either in-person or digitally, contingent on the patient’s condition. Subsequent to the analysis of the results, the anesthesiologist collaborates with the patient to determine the most suitable anesthesia technique. It is worth emphasizing that our ambulatory spine center is structured to not only provide an environment conducive for safe spine surgeries but also to prioritize patient education and individualized care, in strict alignment with the JCI standards.

### 2.5. Anesthesia

Anesthesia undeniably plays a pivotal role in the success of ambulatory spine surgery. At our center, prior to any surgical intervention, a rigorous pre-operative testing and risk assessment is undertaken for every patient. This comprehensive assessment evaluates the patient’s overall health status, medical history, potential allergies, and other critical factors that may influence the choice of anesthesia and the surgery itself, such as cardiac or endocrinological conditions. One week before the scheduled procedure, each patient undergoes this extensive assessment and any required tests. These may include blood work, ECG, echocardiograms, consultations, and other diagnostic investigations to ascertain their suitability and safety for anesthesia.

The anesthesiologist, after gathering these comprehensive data, then sits down with the patient to discuss the potential anesthesia options, along with their benefits, risks, and implications. Together, they weigh the pros and cons of each technique in light of the patient’s specific health status and procedure. The patient is encouraged to ask questions, express concerns, and share their preferences. In this way, both the anesthesiologist’s expertise and the patient’s autonomy are valued, and a joint decision is made on the most suitable anesthesia technique. This shared decision-making process is paramount to ensure both the patient’s safety and comfort.

Our primary anesthesia modalities include total intravenous anesthesia (TIVA) and regional anesthesia techniques, like epidural anesthesia. While TIVA utilizes intravenous medications to induce and maintain general anesthesia, regional anesthesia is employed to block nerve sensations in specific regions of the body, preventing pain transmission during the surgery. Our meticulous approach to selecting the anesthesia, fortified via thorough pre-operative testing and risk assessment, underscores our commitment to ensuring our patients’ safety and comfort throughout their ambulatory spine surgery experience.

### 2.6. Procedure

Most patients received TIVA, while epidural anesthesia with analgosedation was employed in individual cases. The procedures were performed using microdiscectomy or endoscopic discectomy with the transforaminal or interlaminar approach, depending on the surgeon’s preferences and anatomical conditions. All procedures were performed by one of the five neurosurgeons who are partners of the company running this center. There were a maximum of four procedures in one day of the week, with the last one ending no later than 5 p.m.

### 2.7. Outpatient Management

After the procedure, each patient was monitored in a two-person post-operative room, accompanied by another post-surgery patient to maximize the efficient use of space and resources. A qualified physiotherapist mobilized the patient 1–2 h after the procedure concluded, offering guidance on post-operative behavior, potential exercises, and recommended physical therapy treatments. Critical post-operative metrics, such as the level of pain experienced, analgesic requirements, voiding status, and mobility status, were meticulously documented by the anesthesiology team.

### 2.8. Surgical Follow-Up

Upon their discharge, patients were provided with direct telephone contact to both their surgeon and anesthesiologist, a measure put in place to ensure an uninterrupted line of communication for any post-operative concerns or queries. The surgeon initiated the first control phone call on the morning following the surgery, and this was not merely a one-time check-in. It was the start of a structured monitoring process, where any signs of complications, deviations from an expected recovery trajectory, or patient-reported concerns triggered a more intensive follow-up. This could range from additional phone calls, early in-person check-ins, or consultations with other specialists, depending on the concern. The subsequent check-in call a week later aimed at tracking the patient’s progress, ensuring that their pain management was effective, and that there were no signs of complications. Throughout this period, any red flags, such as increased pain, signs of infection, or concerns about mobility, would have prompted immediate action. The standard in-person follow-up was set for four weeks post-surgery, but a structured list of criteria was set in place, and any meeting of these criteria led to an earlier appointment. This iterative and responsive approach, coupled with specific criteria for intervention, ensured that our monitoring of patients post-surgery was both close and effective. The visual analog scale (VAS) was utilized both before the surgical intervention and one week post-surgery. This evaluation was conducted using an online questionnaire. Similarly, the core outcome measurement index (COMI), which had previously been validated in the Polish language in 2013 [8], was measured both pre-operatively and 12 months post-operatively using a dedicated online questionnaire. At the 12-month mark, an additional question was posed to patients, inquiring whether they would consider undergoing the same type of procedure in a similar mode again. All these data were promptly incorporated into the Polish Spine Surgery Registry, ensuring a comprehensive and up-to-date record of patient experiences and outcomes.

### 2.9. Statistical Analysis

The statistical analysis of the conducted research was extensive and multifaceted. All possible patient data were carefully checked for various relationships and information. The collected data were analyzed using Matlab R2020b numerical software and its statistical tools. Descriptive statistics, including means and standard errors for continuous variables, were used to summarize most of the collected data. These statistics were helpful in describing datasets on VAS indicators for individual surgical methods, as well as the COMI indicator and return-to-work time. For categorical variables, such as the percentage distribution of the occurrence of given operation methods, a percentage presentation was used for effective data visualization. This allowed for a better representation and understanding of certain facts.

In order to compute the complication risk percentage, we divided the number of patients who encountered complications by the total number of surgeries performed, and then multiplied the result by 100.

## 3. Results

### 3.1. Patient Characteristics

During the three-year study period, a total of 318 outpatient spine surgeries were performed at the ambulatory spine center. These included 95 microdiscectomies (MLDs), 122 endoscopic discectomies (PELDs)—82 transforaminal (TELD) and 40 interlaminar (IELD), 84 anterior cervical discectomies with fusion (ACDFs), 11 posterior endoscopic cervical discectomies (PECDs), and 6 endoscopic interbody fusions with percutaneous screw fixation (endoLIFs) (Figure 2).

Patients between 50 years old and 60 years old were the largest group, accounting for 28% (89) of the total number of procedures. The second-largest group were patients between 40 years old and 50 years old, accounting for 27% (86). Patients under 40 years of age accounted for 18% (56) of the total number of procedures, while patients between 60 years old and 70 years old accounted for 23% (74), and patients over 70 years old accounted for 11% (35) (Figure 3). The average age of patients undergoing outpatient spine surgery was 52.9 years, with the youngest patient being 20 years old and the oldest being 86 years old. In total, 176 patients were male and 142 were female.

### 3.2. Short-Term Follow-Up

The mean pre-operative VAS score was 5.82 (±0.41), which significantly decreased to 1.57 (±0.33) one week post-surgery.

In the microdiscectomy group, the pre-operative VAS score was 6.45 (±0.57), which significantly decreased to 1.52 (±0.53) one week post-surgery.In the ACDF group, the pre-operative VAS score was 4.28 (±0.76), which significantly decreased to 1.11 (±0.59) one week post-surgery.In the endoLIF group, the pre-operative VAS score was 6.00 (±0.82), which significantly decreased to 2.00 (±1.41) one week post-surgery.In the endoscopic posterior cervical discectomy (PECD) group, the pre-operative VAS score was 6.83 (±1.09), which significantly decreased to 2.33 (±1.12) one week post-surgery.In the interlaminar endoscopic system (IELD) group, the pre-operative VAS score was 6.19 (±1.36), which significantly decreased to 2.74 (±1.46) one week post-surgery.In the transforaminal endoscopic spinal system (TELD) group, the pre-operative VAS score was 6.28 (± 0.91), which significantly decreased to 1.57 (±0.66) one week post-surgery (Table 1).

### 3.3. Long Term Follow-Up

Among all patients, the mean time from surgery to recovery or return to work was 3.85 (±0.36) months.

The pre-operative VAS score after 6 months of hindsight was 6.77 (±0.40), with variations observed in different procedure groups (Table 1). It is important to note that the pre-operative VAS score at 6 months of hindsight was higher than the VAS score before surgery, indicating that patients tended to overestimate their symptoms after some time (Table 1).

Long-term follow-up for all patients was assessed using the core outcome measurement index (COMI). This metric was used to evaluate the effectiveness of outpatient spine surgery in improving patients’ quality of life. The pre-operative data showed a mean COMI score of 6.9303 (±0.5690), which significantly improved to a mean post-operative score of 3.2803 (±0.7101) twelve months following the operation (Table 2).

The follow-up rate at 12 months was 78%. The reduced follow-up was primarily due to the lack of contact with the remaining patients: they did not respond to email-based questionnaires and did not answer our phone calls, limiting our ability to collect data from them. Nonetheless, the data we were able to obtain allowed for robust analysis in the long-term assessment of these outpatient spine surgeries. An analysis of the long-term COMI scores across the different procedure groups revealed no statistically significant differences. After 12 months, 91% of patients reported that they would undergo the same procedure again in an outpatient setting if required.

### 3.4. Anesthesia

Of the total patients, 237 were in the ASA 1 group, 78 in the ASA 2 group, and 3 in the ASA 3 group. General anesthesia was employed in 312 cases (98.11%), while local anesthesia was administered in 6 cases (1.89%). Among the patients who received local anesthesia, two underwent MLD, three TELD and one underwent the IELD procedure.

### 3.5. Surgery Time and Post-Operative Care

It was important to clarify the metrics we used in our study. The “time of surgery” referred to the duration of the procedure itself, specifically measured from the initial skin incision to the application of the final dressing, often termed as ‘skin-to-skin’ time. The “time to patient’s verticalization post-operation” was counted from the end of anesthesia to the moment when the patient was mobilized, indicating how soon after the surgery and anesthesia the patient was able to become upright again. Lastly, the “observation time” represented the duration from the completion of anesthesia to the point of patient discharge from our facility. This measured how long the patient remained under our care post-procedure, prior to being released to recover at home.

For the TELD procedure, the mean operation time was 78.35 min (±5.55), the time to patient’s verticalization post-operation was 64.15 min (±6.35), and the patient observation time post-anesthesia was 185.20 min (±6.67).The IELD procedure had a mean operation time of 81.15 min (±10.93), time to patient’s verticalization post-operation of 63.54 min (±5.37), and patient observation time post-anesthesia of 184.70 min (±6.27).For the MLD procedure, the mean operation time was 56.26 min (±4.55), the time to patient’s verticalization post-operation was 66.25 min (±3.77), and the patient observation time post-anesthesia was 200.20 min (±8.29).The PECD procedure had a mean operation time of 80 min (±41.63), time to patient’s verticalization post-operation of 61.25 min (±1.55), and patient observation time post-anesthesia of 220.20 min (±40.67).The ACDF procedure had a mean operation time of 63.73 min (±5.59), time to patient’s verticalization post-operation of 64.26 min (±3.10), and patient observation time post-anesthesia of 364.2 min (±5.76).Finally, the endoLIF procedure had the longest mean operation time of 183.33 min (±101.71), time to patient’s verticalization post-operation of 85.14 min (±3.10), and patient observation time post-anesthesia of 390.21 min (±66.79) (Table 3).

### 3.6. Complications and Adverse Effects

Over the span of three years, our center conducted more than 300 spinal procedures under general anesthesia, of which 227 were on the lumbar spine. We encountered two infectious complications: a superficial wound infection and spondylodiscitis. Incidental durotomy occurred in two cases, leading to intra-operative cerebrospinal fluid leakage. Fortunately, these instances had no subsequent clinical repercussions and did not necessitate longer patient follow-ups.

Another notable case involved a patient, post-epidural anesthesia, who experienced urinary retention. Further examination linked this to a previously undetected severe prostatic hyperplasia. Post-operatively, four patients faced recurrent herniation, leading to subsequent surgeries either within our facility or at a hospital by the same surgical team. Additionally, a case of spondylodiscitis led a patient to seek screw fixation at another facility.

We had two instances that necessitated patient transfers to the hospital neurosurgery department: one due to post-procedural neurological decline (which then called for a more extensive surgical intervention at the hospital), and another due to sustained post-anesthetic confusion. Following these cases, we undertook an exhaustive review of the surgical and post-operative procedures, alongside cross-referencing patient histories and intra-operative events. Despite our meticulous assessments, the specific factors contributing to these adverse outcomes remained unidentified. Both procedures adhered to the best practice guidelines and were conducted without deviations, as corroborated through internal audits and peer reviews (refer to Table 4).

Interestingly, throughout our operations, no records indicated increased post-operative pain or an extended period of immobilization that may have impeded a patient’s same-day discharge.

## 4. Discussion

Outpatient spine surgery is an emerging trend in healthcare, particularly in the fields of urology, aesthetic surgery, ophthalmology, orthopedics, and, increasingly, spine surgery [9]. The treatment of spinal diseases is one of the largest financial burdens for the global healthcare system, with treatment costs reaching tens of billions of dollars annually [10]. Considering the current global economic crisis and ongoing conflicts, the financial burden on the healthcare system is particularly high [15]. Therefore, it is crucial to minimize unnecessary costs related to prolonged hospitalization. Outpatient spine surgery appears to be an ideal solution to cut costs without compromising patient outcomes.

The literature has revealed that outpatient surgery can provide similar clinical efficacy compared to traditional surgery and, in some cases, even better outcomes [12,16,17,18]. However, one of the primary concerns associated with outpatient surgery is the risk of post-operative pain and late post-operative or anesthetic complications [19]. Although outpatient procedures have a lower risk of complications, it is important to note that severe complications can still occur, and outpatient facilities must be adequately equipped and prepared to manage them.

In the group of patients we treated, only two out of three hundred required hospital transfer—one due to surgical complications and the other because of issues related to anesthesia. This rate of transfer is encouraging when compared to the broader literature on the subject [20,21]. Several studies have indicated that ambulatory or outpatient procedures often come with a reduced risk of complications [22]. However, it is important to put this into context: Many patients who undergo outpatient surgeries tend to be younger, without significant co-morbidities, and are generally in good health. Such a demographic naturally tends to have fewer complications. This factor may be influencing the overall lower complication rates seen in outpatient settings, rather than it being solely a result of the procedure type or setting itself.

In our cohort, it is worth noting that three patients were classified into the ASA 3 category. For such patients to be considered for outpatient procedures, the underlying reason for their ASA 3 classification typically stems from a chronic disease that is currently in a stable phase and is being optimally managed. This distinction is vital, as it underlines the fact that their health status, though not perfect, is not acutely volatile either. Furthermore, in our center, these particular patients underwent regional anesthesia, as it was deemed a safer option for them, considering their health background and the nature of their chronic conditions. Additionally, for these patients, we employed the least invasive technique, endoscopic discectomy, to further ensure safety and optimize the post-operative recovery process.

The use of minimally invasive surgical techniques, such as endoscopy, has further reduced the risk of complications in outpatient procedures, including spine surgery. Approximately one-third of our patients underwent endoscopy, and only one case of recurrent hernia was observed in this group [23,24]. Endoscopic procedures have been shown to have a lower risk of complications and a similar risk of recurrent hernia compared to microdiscectomy, and patients who undergo endoscopic procedures have a faster return to work, which has significant economic importance [25,26].

The use of endoscopic techniques has greatly expanded the possibilities for minimally invasive outpatient spine surgery, allowing for procedures such as posterior cervical discectomy (PECD) and endoscopic interbody fusion with percutaneous screw fixation (endoLIF) [27,28]. These techniques provide many advantages over traditional open surgery, including smaller incisions, reduced blood loss, and shorter hospital stays [28,29]. In particular, the use of endoscopic stabilization techniques offers new possibilities for outpatient spine surgery, enabling spine stabilization procedures with rapid mobilization and discharge within a few hours [29,30]. This is something that was previously impossible due to the significant post-operative pain associated with traditional surgery, which is either eliminated or greatly reduced with the employment of endoscopic techniques [30]. Overall, the availability of these endoscopic procedures has significantly improved the clinical effectiveness and safety of outpatient spine surgery, allowing for a more efficient and patient-friendly approach to the treatment of spinal disorders.

Our center also successfully performed endoscopic interbody fusion with percutaneous screw fixation (endoLIF) in six cases, which is another example of the potential benefits of endoscopic techniques in ambulatory spine surgery [31,32]. This procedure, which involves the use of an endoscope to access and stabilize the spine, has been shown to be a safe and effective option for certain patients. Similarly, we performed posterior cervical discectomy (PECD) in 11 cases with no complications, demonstrating the potential for endoscopic techniques to expand the range of outpatient spine surgery procedures [33,34]. These results highlight the increasing importance of endoscopic techniques in the field of ambulatory spine surgery, and their potential to provide safe, effective, and minimally invasive treatments for a variety of spinal disorders.

While outpatient surgeries have a lower risk of complications, severe complications, such as bleeding, nerve damage, or infection, can still occur [35,36,37,38,39]. In the event of a complication, the facility must be equipped with the necessary tools and resources to manage the situation quickly and effectively. This includes having a trained team of medical personnel who are experienced in managing these types of complications and the proper equipment and medication readily available [38].

In addition, proper patient selection is crucial in outpatient spine surgery. Patients who have significant comorbidities or who are at higher risk for complications should be carefully evaluated and may be better suited for traditional hospital-based surgery [37,38]. Close communication and cooperation between the surgeon and the anesthesiologist during patient selection and procedure planning are essential in reducing the risk of complications.

Ensuring patient safety is paramount, and outpatient facilities bear the responsibility of meeting all the essential regulatory requirements and standards. This not only encompasses adherence to both local and national regulations but also mandates the establishment of the appropriate policies and procedures to adeptly manage potential complications.

As we mentioned earlier, proper patient selection is crucial for outpatient spine surgery. However, our patient population reflects the epidemiology of spinal diseases [39], which is increasingly affecting older individuals. In our group, patients over 70 years old accounted for nearly 11% of the total number of procedures performed. This is in line with other studies reporting that the prevalence of spinal diseases increases with age, particularly in those over 65 years old [40]. While some may consider advanced age a contraindication for outpatient spine surgery, our results demonstrate that carefully selected elderly patients can safely undergo these procedures in an ambulatory setting. Nonetheless, the decision to perform outpatient surgery should always be made on an individual basis, taking into account the patient’s overall health and comorbidities [38].

In our analysis of surgery times, patient mobilization, and the duration of stay at the facility, we observed specific patterns consistent with those reported by other authors, such as Helseth et al. in their study of 1449 outpatient spine patients [41]. Most procedures were performed within a comparable time frame, averaging around one hour. Endoscopic procedures tend to take slightly longer, likely due to the necessity of preparing and handling a larger amount of equipment, such as fiber-optic cameras, water drainage pumps, etc., all of which are included in this time frame. The stabilization procedures were found to take significantly longer, although it should be noted that these were the team’s first of this kind, and a learning curve must be considered. Concerning the time of stay and patient mobilization, our facility adheres to consistent guidelines, mirroring those found in similar studies. All patients are mobilized within 1–2 h post-surgery. For lumbar spine procedures, patients do not leave the facility earlier than 3 h, and for cervical spine procedures, the period extends to 6 h. These strategies align with a broader trend towards efficiency and safety in outpatient spine surgery, further substantiating the feasibility of these methods under various surgical contexts.

In our analysis, we placed significant emphasis on the clinical outcomes post-surgery, observing a notable decline in both the VAS and COMI scores. Our findings regarding the improvement in clinical conditions align with the existing literature. However, as astutely pointed out by Sivaganesan in his review [42], the majority of these literature reports, including ours, are based on retrospective analyses. Despite their promising results, they do not conclusively prove the safety and efficacy of these procedures. This underscores the need for further, more controlled studies to definitively ascertain the safety and effectiveness of ambulatory spine surgeries.

In reviewing the available literature, it was challenging to find articles highlighting the negative aspects of ambulatory spine surgery. While many reports, including ours, portray favorable outcomes, this predominance of positive results might suggest either a genuine benefit of these procedures or a publication bias favoring positive outcomes. Regardless, it emphasizes the necessity for comprehensive, unbiased research to provide a balanced perspective on the pros and cons of ambulatory spine surgeries.

In another study of ours, we examined the prevalence of outpatient spine procedures in Poland [43]. One of the primary reasons given by Polish spine surgeons for not adopting outpatient procedures was their apprehension about potential complications and the associated legal liabilities. Additionally, organizational challenges in their respective centers also contributed to their hesitancy.

Outpatient spine surgery in Poland, though not yet pervasive, presents a cost-effective method for patient treatment. This efficiency is particularly prominent when harnessing contemporary anesthetic techniques alongside minimally invasive surgical procedures. Our analysis revealed that a range of foundational spine surgeries, such as microdiscectomy, endoscopic discectomy, ACDF, posterior endoscopic cervical discectomy, and short spine stabilization, can be securely executed within ambulatory spine centers. Nevertheless, for paramount patient safety, it is essential to delineate both the specific surgical procedures and the requisite equipment vital for managing grave complications.

Certainly, financial comparisons across healthcare settings can be intricate due to regional disparities and unique administrative frameworks, such as the one in Poland. Nevertheless, based on direct procedure costs, our facility demonstrates cost-effectiveness when juxtaposed against a neighboring public hospital. This notion was further bolstered by studies from other regions, notably a U.S.-based research suggesting that outpatient spine surgery can diminish healthcare expenditures by as much as 30% when contrasted against inpatient surgical procedures [26,44].

Furthermore, and it is crucial to approach this with utmost caution and objectivity, leveraging outpatient centers may help alleviate some strain on overwhelmed hospital systems. The idea is not to underplay the vital role hospitals play but to highlight potential strategies to optimize resource allocation, especially during healthcare emergencies. With the ongoing challenges posed by events such as the COVID-19 pandemic, and other unforeseen calamities like wars or natural disasters, a strategic division of tasks could enhance healthcare responsiveness. By managing certain procedures in outpatient settings, hospitals might be better positioned to cater to patients demanding acute care and longer hospitalizations.

### Limitations and Scope of the Study

Our study is a retrospective analysis of cases treated at our center, without a control group for comparison. The goal of our study was not to assert that outpatient treatments are superior but to shed light on the potential and capabilities of outpatient procedures given the specific patient qualifications. We understand the importance of rigorous and objective analyses, and our emphasis was on presenting our findings within the context of our facility and patient population. Our aims were to show the broad spectrum of procedures that can currently be performed in an outpatient setting and to outline the operational model adopted at our center. From our perspective, these aspects are extremely important, especially considering that in our country and region, such procedures are extremely rare and there are no legal regulations sanctioning the operation of similar facilities. Despite these constraints, we believe that the insights and experiences shared in this study can serve as valuable points of reference for other institutions seeking to explore or expand their capabilities in outpatient spine surgery.

Looking forward, we advocate for further research in this area to continue improving patient outcomes and expanding the range of procedures that can be performed under an outpatient setting. We also call for the development and implementation of legal regulations that would sanction and standardize the operation of similar facilities to ensure the safety and efficacy of outpatient spine surgery.

## 5. Conclusions

In our study, we explored the potential benefits and challenges of outpatient spine surgery. There was a marked decrease in the VAS and COMI scores post-surgery, signaling substantial clinical improvements among the patients. An overwhelming majority of the patients expressed that they would willingly undergo the procedure in an ambulatory setting again, attesting to the perceived efficacy and positive experience of the procedure.

While our findings suggest that, under certain conditions, outpatient spine surgery might be a viable option for specific procedures, such as microdiscectomy, endoscopic discectomy, ACDF, posterior endoscopic cervical discectomy, and short spine stabilization, it is important to highlight that our analysis was based on retrospective data from our center without a control group.

Furthermore, our work largely illustrates the technical aspect of how an ambulatory spine surgery center might be organized and operated in Poland. It is noteworthy to mention that, to date, our center stands as the sole establishment of its kind in the country. Within our patient group, we achieved satisfactory clinical outcomes with, in our opinion, a low complication rate. The observed outcomes regarding patient recovery, complications, and cost considerations point towards favorable trends for outpatient procedures. However, these conclusions should be interpreted with caution. Comprehensive patient selection, the use of minimally invasive techniques, and collaboration between the surgical and anesthesia teams were emphasized as key components for success.

While we believe that our research provides valuable insights into the potential of outpatient spine surgery in Poland, it is evident that further research, possibly involving broader and more diverse datasets and control groups, is necessary to derive more definitive conclusions.

## Figures and Tables

**Figure 1 healthcare-11-02944-f001:**
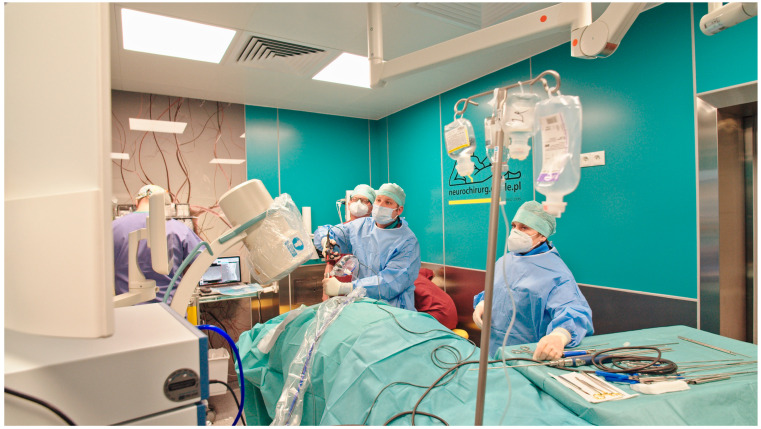
Real-time view of the treatment room during an endoscopic spinal surgery.

**Figure 2 healthcare-11-02944-f002:**
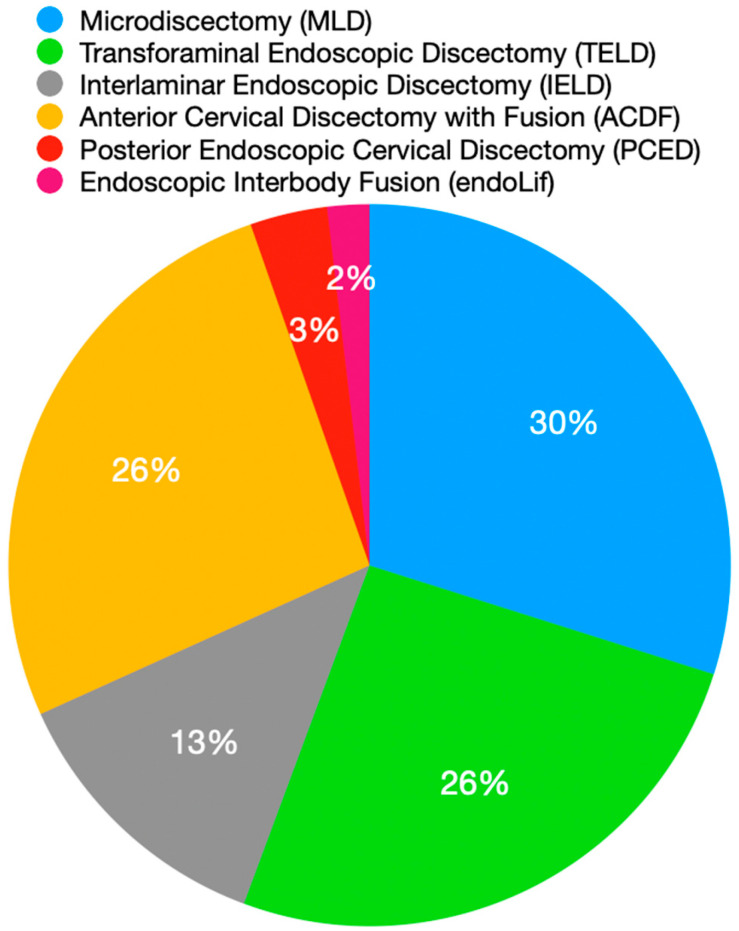
Distribution of procedures performed in the outpatient spine surgery center between 2018 and 2021.

**Figure 3 healthcare-11-02944-f003:**
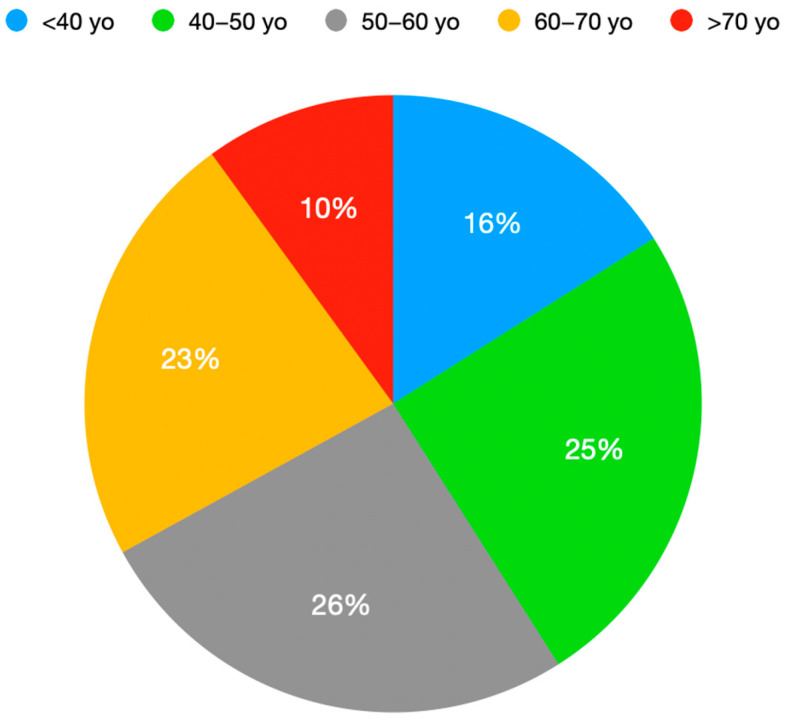
Age distribution of patients (*n* = 318) undergoing outpatient spine surgery.

**Table 1 healthcare-11-02944-t001:** Pre-operative and post-operative visual analog scale scores for various spine procedures.

Procedure Group	Pre-Operative VAS	VAS One Week after Surgery	Pre-Operative VAS Declared after Six Months
MLD	6.45 (SD = 0.57)	1.52 (SD = 0.53)	7.02 (SD = 0.48)
ACDF	4.28 (SD = 0.76)	1.11 (SD = 0.59)	4.76 (SD = 0.88)
endoLIF	6.00 (SD = 0.82)	2.00 (SD = 1.41)	5.00 (SD = 1.73)
PECD	6.83 (SD = 1.09)	2.33 (SD = 1.12)	6.83 (SD = 1.09)
IELD	6.19 (SD = 1.36)	2.74 (SD = 1.46)	8.30 (SD = 1.16)
TELD	6.28 (SD = 0.91)	1.57 (SD = 0.66)	8.00 (SD = 0.59)

Note: VAS score = visual analog scale score, ACDF = anterior cervical discectomy with fusion, endoLIF = endoscopic interbody fusion with percutaneous screw fixation, PECD = endoscopic posterior cervical discectomy, IELD = interlaminar endoscopic lumbar discectomy, TELD = transforaminal endoscopic lumbar discectomy.

**Table 2 healthcare-11-02944-t002:** Comparison of pre-operative and long-term outcomes of minimally invasive lumbar microdiscectomy using the COMI scale.

	Pre-Operative COMI	Twelve Months Follow-Up COMI
Long-term follow-up	6.93 (SD = 0.57)	3.28 (SD = 0.71)
Mean return-to-work time	3.85 (SD = 0.36) months	

Note: COMI = core outcome measurement index.

**Table 3 healthcare-11-02944-t003:** Comparison of operation times, time to patient verticalization post-operation, and patient observation time post-anesthesia across different procedures.

Procedure	Mean Operation Time (min)	Time to Patient Verticalization Post-Operation (min)	Patient Observation Time Post-Anesthesia (min)
TELD	78.35 (SD = 5.55)	64.15 (SD = 6.35)	185.20 (SD = 6.67)
IELD	81.15 (SD = 10.93)	63.54 (SD = 5.37)	184.70 (SD = 6.27)
MLD	56.26 (SD = 4.55)	66.25 (SD = 3.77)	200.20 (SD = 8.29)
PECD	80.33 (SD = 41.63)	61.25 (SD = 1.55)	220.12 (SD = 40.67)
ACDF	63.73 (SD = 5.59)	64.26 (SD = 3.10)	364.2 (SD = 5.76)
endoLIF	183.33 (SD = 101.71)	85.14 (SD = 3.10)	390.21 (SD = 66.79)

Note: VAS score = visual analog scale score, ACDF = anterior cervical discectomy with fusion, endoLIF = endoscopic interbody fusion with percutaneous screw fixation, PECD = endoscopic posterior cervical discectomy, IELD = interlaminar endoscopic lumbar discectomy, TELD = transforaminal endoscopic lumbar discectomy.

**Table 4 healthcare-11-02944-t004:** Complications and undesirable effects of ambulatory spine surgery.

Complications and Undesirable Effects	No. of Cases	Reoperation	Direct Hospital Transfer	Complication Risk
Dural tear	2	0	0	0.63%
Superficial infection	1	0	0	0.31%
Spondylodiscitis	1	1	0	0.31%
Recurrent hernia	4	2	0	1.26%
Neurological deterioration	1	1	1	0.31%
Urinary arrest	1	0	0	0.31%
Anesthesia complications	1	0	1	0.31%
Total	9	4	2	2.83%

## Data Availability

The data presented in this study are available on request from the corresponding author. The data are not publicly available due to privacy reasons.

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
