# Peer review of "Outpatient Spine Procedures in Poland: Clinical Outcomes, Safety, Complications, and Technical Insights into an Ambulatory Spine Surgery Center"

_healthcare, 2023, doi:10.3390/healthcare11222944_

Round 1
Reviewer 1 Report (Previous Reviewer 1)
Comments and Suggestions for Authors
The introduction should be linked to the aim of the study (effective).
The study's conclusions are not supported by the results presented. What does safety have to do with the VAS score and COMI-back? How are they related?
"The primary goal of this study was to gauge the safety, efficacy, and levels of patient satisfaction related to outpatient spine surgeries in an ambulatory environment." How? The data collection instruments do not meet these goals. The Core Outcome Measures Index for the back (COMI-back) is a instrument for assessing the main outcomes of importance to patients with back problems (pain, function, symptom-specific well-being, quality of life, disability).
"This article aims to increase awareness of the advantages of ambulatory spine surgery and encourage its adoption in the Polish healthcare system." How is this the aim of the study? Isn't the aim to evaluate the surgical outcomes assessed using the aforementioned scales?

Author Response
Dear Reviewer,
Thank you for your thoughtful feedback and questions regarding our study. Please allow us to address the concerns you've raised:
-
Safety Concerns Linked with VAS and COMI-back:
- We'd like to emphasize that while the VAS and COMI-back scales were central tools for assessing clinical outcomes, they were not the only metrics we analyzed in the context of safety. The frequency of complications, reoperations, and instances where a patient required transfer to a hospital were also meticulously examined. Moreover, we detailed our operational model, which, in our opinion, enhances the safety of such procedures.
-
Purpose of COMI and VAS Tools:
- You're right in noting that the COMI-back instrument evaluates primary outcomes significant to patients with back problems, including pain, function, symptom-specific well-being, quality of life, and disability. However, our research didn't solely rely on the VAS and COMI for all our objectives. Patient satisfaction was also ascertained through an additional question that inquired whether they would opt for an ambulatory procedure again. This, in our view, also provides insight into the perceived efficacy and safety from a patient's perspective. The safety aspects, as previously mentioned, were addressed more comprehensively than solely through these scales.
-
Aim of the Study:
- Our primary objective is indeed to promote ambulatory solutions within the Polish healthcare system. While clinical evaluation and demonstrating tangible improvements for patients are critical, they serve as tools or evidence supporting our larger aim. By showcasing the positive outcomes and safety of ambulatory spine surgeries through these scales and other metrics, we hope to underscore the advantages and potential of this approach for a broader adoption in the Polish healthcare setting.
Furthermore, in response to your feedback and to ensure the accuracy and clarity of our work, we've made necessary corrections in the results section, refining our data presentation and adding supplementary analysis where needed.
We trust that these clarifications address your concerns and provide a clearer perspective on our study's rationale and methodology. We sincerely value your feedback as it ensures our research is robust, well-articulated, and meaningful to the field.
Warm regards,
Kajetan Latka MD PhD
Reviewer 2 Report (Previous Reviewer 2)
Comments and Suggestions for Authors
I reviewed the article, and it seems quite good enough in the point of my view.
I would accept the article with the authors' correction.
Author Response
Dear Reviewer,
Thank you very much for taking the time to review our manuscript and for your positive feedback. We sincerely appreciate your constructive remarks and are grateful for the guidance you provided.
We have diligently made the corrections as suggested, ensuring the integrity and quality of our work. Your endorsement of our article and the recognition of its merits is truly encouraging.
We believe that collaborative efforts, such as this peer-review process, significantly contribute to advancing the quality of scientific literature. Once again, we extend our gratitude for your contribution to this endeavor.
Warm regards,
Kajetan Latka MD PhD
This manuscript is a resubmission of an earlier submission. The following is a list of the peer review reports and author responses from that submission.
Round 1
Reviewer 1 Report
Comments and Suggestions for Authors
Comments on the Quality of English LanguageIt is necessary to clarify which research question(s) the authors address and which methods they use to answer the question. The rationale of the study is unclear. Longitudinal or retrospective?
From the contents, the description of certain indicators is considered, but there is no way of supporting certain results.
Reservations regarding the analysis and interpretation of the results, as well as their contribution to the scientific field.
Author Response
Thank you for your constructive feedback and concerns regarding our manuscript.
-
Objective and Integrity: We want to emphatically clarify that our primary motivation in presenting our findings is academic and informative. The intent is to contribute to the broader understanding of outpatient spine surgery, share our experiences, methodologies, and insights, and potentially guide similar medical establishments in refining their procedures. With the limited prevalence of such surgeries in our region, we believed that detailing our process could provide valuable insights to the global medical community.
-
Revisions and Updates: Based on your feedback, we have made necessary corrections to the highlighted paragraphs, including a thorough revision of our methodology section. We've provided a more precise outline of the criteria qualifying and disqualifying patients from ambulatory procedures. Additionally, we've clarified the organizational and legal aspects of our outpatient facility's functioning. We've also revisited our conclusions and summary sections to ensure they are not overly presumptuous.
-
Focus on Legal Regulations: Our emphasis on the pressing need for legal regulations is an affirmation of our commitment to prioritizing patient safety and advocating for standardized care protocols. This advocacy is intended to highlight broader issues in our field, rather than merely promoting our specific center.
-
Transparency and Patient-Centric Focus: We fully concur with the necessity of transparent and unbiased communication of advancements in medical science. Our dedication remains unwavering towards patient care, and this manuscript seeks to serve that purpose by fostering a platform for discourse on best practices in outpatient spine surgery.
Your vigilance in ensuring the integrity of research is highly valued. We hope the revisions we have made address your concerns adequately, emphasizing our commitment to the betterment of medical practices and patient outcomes.
Warm regards
Reviewer 2 Report
Comments and Suggestions for Authors
Interesting topic and glad to see the good results on spine surgery.
In the methods, how n when to collect VAS and COMI must be addressed.
There are some fragment paragraph, and they need to be rearranged.
The lines of table must be corrected as research table.

Author Response
Thank you for your kind words regarding our topic and the results of our study on outpatient spine surgery. We greatly appreciate the time and effort you've invested in reviewing our manuscript and providing valuable feedback.
-
VAS and COMI Collection: You correctly pointed out the need to provide a clearer description of the VAS and COMI data collection methodology in the methods section. We have now included detailed information on the timing and procedure of collecting these scores. Specifically, VAS analysis was conducted before the procedure and one week post-operation using an online questionnaire. Similarly, the COMI score was measured pre-operatively and 12 months post-operation through an online questionnaire. Upon your recommendation, we have added a comprehensive discussion section addressing our results concerning VAS and COMI scores. This section compares our findings with relevant literature and delves deeper into the implications of our results in the broader context of spine surgery outcomes.
-
Fragmented Paragraphs: Thank you for bringing this to our attention. Based on your feedback, we carefully reviewed the manuscript and have restructured and rearranged the fragmented paragraphs to improve the flow and clarity of the content.
-
Table Formatting: We sincerely apologize for the oversight. The tables have now been reformatted and corrected to align with standard research table formats. We hope this enhances the readability and comprehension of the data presented.
Your insights have been instrumental in refining our manuscript and ensuring that our work aligns with the highest academic standards. We believe that the revisions have enhanced the paper's quality and addressed the concerns raised.
We are grateful for your constructive critique and look forward to any further recommendations or feedback you might have.
Warm regards
Round 2
Reviewer 1 Report
Comments and Suggestions for Authors'We rigorously analyze complications every six months in a collaborative setting, involving our entire medical team in discussions, deliberations, and decision-making processes to enhance our practices and protocols.' How do you identify the research design as retrospective? If you evaluated the VAS pre and postoperatively, and operationalized the COMI, how is it retrospective? Is this assessment registered? It is unclear how we share this retrospective study.
How do you evaluate surgical effectiveness using these two instruments?
'This article aims to increase awareness of the advantages of ambulatory spine surgery and encourage its adoption in the Polish healthcare system.' How does the aim of this article tie in with the aim of the research? How does a retrospective study respond to this aim?
'While our center's treatment room is not legally classified as a surgical operating room, it is registered and functions as a treatment room.' How is it possible to perform surgeries in a place that is not legally accepted to perform them?
'The Visual Analog Scale (VAS) was utilized both before the surgical intervention and one week post-surgery. This evaluation was conducted using an online questionnaire. Similarly, the core outcome measurement index (COMI) was measured both pre-operatively and 12 months post-operatively using a dedicated online questionnaire. At the 12-month mark, an additional question was posed to patients, inquiring if they would consider undergoing the (...)'. How do you consider this retrospective?

Author Response
- To clarify, our study wasn't originally designed as a prospective investigation. Instead, the evaluation of VAS and COMI pre- and post-operatively forms a part of our standard practice aimed at monitoring the well-being and progress of our patients. As such, this practice has been routinely instituted irrespective of this specific study. When we decided to analyze and share our findings for this particular study, we utilized the data that had already been collected as a part of our routine patient monitoring, which inherently gives our analysis a retrospective nature.
- Our primary intent with this article is to elucidate the operational mechanism of our unit within the specific context of the Polish healthcare landscape. By shedding light on our practices and processes, we hope to promote a similar work model in other institutions or settings. The retrospective analysis of questionnaire outcomes serves to underscore that, using this model, we achieve satisfactory clinical improvements in our patients. It is a testament to the efficacy of our processes, even as it complements the primary focus on operational mechanisms.
- We apologize for any confusion or ambiguity our previous wording may have caused. We'd like to clarify that we never stated our treatment room is unauthorized for procedures. Rather, we emphasized that it isn't registered as a "surgical operating room" but functions as a "treatment room." Importantly, the Regional Sanitary and Epidemiological Station has approved the use of our treatment room for performing specific minimally invasive spine surgery procedures. We have revised the text to clearly reflect this distinction, and we trust this provides the necessary clarity on the matter.
- Thank you for pointing out the potential discrepancy. To clarify, while the collection of VAS and COMI scores was a systematic and prospective aspect of our patient monitoring, the design of our study is retrospective in nature. We did not initiate the study with a predefined hypothesis or plan for collecting these data for research purposes. Instead, these scores are routinely collected as part of our standard patient care protocol. The data was retrospectively analyzed for the purposes of this research. Our intent was to evaluate the outcomes and effectiveness of the procedures post-factum, utilizing the already gathered data.